# A Multidisciplinary Approach to Patients with Psoriasis and a History of Malignancies or On-Treatment for Solid Tumors: A Narrative Literature Review

**DOI:** 10.3390/ijms242417540

**Published:** 2023-12-16

**Authors:** Nerina Denaro, Gianluca Nazzaro, Giulia Murgia, Federica Scarfì, Carolina Cauchi, Carlo Giovanni Carrera, Angelo Cattaneo, Cinzia Solinas, Mario Scartozzi, Angelo Valerio Marzano, Ornella Garrone, Emanuela Passoni

**Affiliations:** 1Oncology Unit, Fondazione IRCCS Ca’ Granda Ospedale Maggiore Policlinico, 20122 Milan, Italy; carolina.cauchi@policlinico.mi.it (C.C.); ornella.garrone@policlinico.mi.it (O.G.); 2Dermatology Unit, Fondazione IRCCS Ca’ Granda Ospedale Maggiore Policlinico, 20122 Milan, Italy; gianluca.nazzaro@policlinico.mi.it (G.N.); giulia.murgia@unimi.it (G.M.); carlo.carrera@policlinico.mi.it (C.G.C.); angelo.cattaneo@policlinico.mi.it (A.C.); angelo.marzano@unimi.it (A.V.M.); emanuela.passoni@policlinico.mi.it (E.P.); 3UOSD Dermatology, USL Toscana Centro-Prato Hospital, 59100 Prato, Italy; scarfif@gmail.com; 4Section of Dermatology, Department of Health Sciences, University of Florence, 50134 Florence, Italy; 5Medical Oncology Department, University of Cagliari, 09042 Cagliari, Italy; czsolinas@gmail.com (C.S.); marioscartozzi@gmail.com (M.S.); 6Department of Pathophysiology and Transplantation, Università degli Studi di Milano, 20122 Milan, Italy

**Keywords:** psoriasis, anti IL-17, anti IL-23, chemotherapy, cancer, neoplasm, immunotherapy

## Abstract

Psoriasis is a chronic immune-mediated disease that is linked to an increased risk of cancer. Although numerous studies have explored whether neoplasms are concurrent conditions or are induced by psoriasis, a definitive definition remains elusive. In this study, we conducted a comprehensive narrative literature review to offer practical guidance to oncologists and dermatologists regarding the initiation and discontinuation of biologics for psoriasis. The findings indicate that a customized approach is recommended for each patient, and that a history of malignancies does not constitute an absolute contraindication for biologics. Growing evidence supports the treatment of selected patients, emphasizing a nuanced assessment of benefits and risks. There is a lack of data specifying a safe timeframe to initiate biologics following a neoplasm diagnosis due to influences from cancer-related and patient-specific characteristics impacting prognosis. Some patients may continue anti-psoriasis therapy during cancer treatments. Enhanced comprehension of the biological mechanisms in cancer progression and the immune microenvironment of psoriasis holds promise for refining therapeutic strategies. In conclusion, a personalized treatment approach necessitates collaboration between oncologists and dermatologists, considering factors such as cancer prognosis, psoriasis clinical manifestations, patient characteristics, and preferences when making treatment decisions.

## 1. Introduction

Psoriasis is a long-term inflammatory disorder that primarily affects the skin, and is characterized by red, scaly patches that cause significant distress, economic burden, and social impact. The majority of psoriasis patients have only cutaneous symptoms, with approximately 20–30% developing psoriatic arthritis (PsA) over their lifespan.

Impaired functioning of the innate immune system sets off a chain reaction, starting with the activation of dendritic cells and culminating in the activation of adaptive immune responses, accompanied by the production of a cascade of cytokines [1].

More than three decades ago, it was evident that lymphocyte infiltrates were correlated with psoriasis, and patients with psoriasis undergoing bone marrow transplantation or treatment with immunosuppressive agents, such as cyclosporine and methotrexate, experienced dramatic improvements in their inflammatory skin lesions [2]. 

Because the inflammatory infiltrate in lesioned skin is composed mainly of CD4+ and CD8+ T cells, selective inhibition of activated T cells in patients through a novel fusion protein composed of human interleukin (IL) IL-2 and diphtheria toxin fragments (DAB389IL-2) provided definitive proof of the pathogenic role of T cells in psoriasis [2].

The IL-23/IL-17 axis plays a central role in psoriasis pathogenesis, but different mechanisms are associated with distinct psoriasis subtypes. Whereas the TNFα–IL-23–Th17 axis plays a central role in T cell-mediated plaque psoriasis, the innate immune system appears to have a more prominent role in the pustular variants of psoriasis [3].

As our comprehension of the immunopathogenesis of psoriasis has advanced, so too has the arsenal of psoriasis treatments. Patients with varying degrees of severity can now be treated with a variety of topical, systemic, and biologic drugs. Refractory psoriasis is frequently treated with biological therapies, which have been shown to be both safe and efficacious [4,5]. Table 1 shows biological treatments approved in moderate-to-severe chronic plaque psoriasis.

According to many authors, psoriatic patients have a higher risk of developing cancer, although the exact relationship between psoriasis and cancer development is unclear and may be influenced by several factors [6,7,8].

The available evidence indicates that psoriasis raises the risk of developing myocardial infarction, stroke, and death caused by cardiovascular diseases. In addition to an increased risk of cardiometabolic disease, psoriasis is also associated with a higher prevalence of gastrointestinal and chronic kidney disease.

Long-term cancer survivors are a growing reality. A history of cancer is a potential comorbidity for many dermatological patients.

Collaboration between oncologists and dermatologists is critical considering increases in patients with a history of cancer and the rising usage of immunomodulatory medications. Pharmacogenomic understanding of biologics and small molecules may help physicians in clinical practice to predict the different responses to treatment [7].

Understanding the relationship between immunomodulation and cancer progression is proving to be difficult, as immunomodulation is becoming recognized as a vital component in managing both psoriasis and cancer progression. Figure 1 depicts the intricate immune system involved in both diseases.

Although the literature only contains case reports and small case series on the use of biological medications in this population, the most recent generation of biological drugs is generally thought to be safe for patients who have previously had cancer.

When making therapeutic decisions, it is advised to take the burden of psoriatic illness and the chance of neoplastic disease progression and recurrence into account.

With a focus on the biological significance of the IL-17 and IL-12/23 families on the tumor microenvironment (TME), we examined the literature on this subject and produced this narrative review.

Furthermore, we believe that we should reply to the following practical clinical questions:Can people suffering from psoriasis who have had cancer in the past receive biologics treatment?Is there a safe time frame?Can we treat psoriasis patients with biologics while undergoing cancer treatment?What is the relationship between anticancer medications and exacerbations of psoriasis?

## 2. Methods

Using the PubMed database, we conducted a study on English-language articles published between 2000 and 2023. A thorough search was conducted using the terms “carcinoma”, “history of malignancies”, and “psoriasis,” along with subheadings such pathophysiology, biologics, anti-TNF, anti-IL-17, anti-IL-23, cancer treatment, and chemotherapy. The following criteria were necessary for inclusion: (1) case series or case reports that addressed the risk of cancer in psoriatic patients, as well as the relationship between biologic psoriasis treatment and the risk of malignancy or cancer recurrence; and (2) review papers, meta-analyses, and systematic reviews centered on the impact of biologics on the risk of cancer in psoriatic patients. Exclusion criteria included (1) papers produced in languages other than English. In addition, chosen pieces from international meetings were used to enhance the discussion of our review.

## 3. Results

### 3.1. Cytokines in Psoriasis and Cancer Microenvironment

Certain biologic drugs used to treat psoriasis may have a pro-tumoral effect, which highlights the immunological mechanisms that are common to both the psoriatic cutaneous environment (PME) and the tumor microenvironment (TME). This imbalance can result in an accumulation of a heterogeneous population of myeloid-derived suppressor cells (MDSCs) and T regulatory T cells (T-reg). A proinflammatory loop in PME causes an overreaction from the immune system, whereas immunosuppressed immune cells and mediators in TME account for the development and metastatization of the disease [8].

In order to avoid immune surveillance, tumor cells employ a variety of strategies, including downregulating tumor antigens or impairing antigen-presenting cells, secreting pro-tumoral cytokines (tumor growth factor (TGF-β), IL-6, vascular endothelial growth factor (VEGF)), losing adhesion molecules and elevated galectin levels, generating epithelial mesenchymal transition, and overexpressing inhibitory immune checkpoints [9].

Psoriatic inflammation in PME is mediated by epidermis-resident memory T cells (Trm) and Langerhans cells, also known as skin macrophages. Trm cells, which produce IL-17 and IL-23, are also present in skin that has previously been psoriasis-affected. It is now thought that these cells are responsible for psoriasis recurrences at the locations of previously healed psoriatic plaques [10].

Wang et al. showed that following biologics, the psoriatic microenvironment varies. Particularly, with treatment, there was evidence of a decrease in infiltrating CD4-Trm, resting NK cells, monocytes, memory B cells, follicular helper T cells, activated DCs, M1 macrophages, and neutrophils and an increase in infiltrating activated NK cells, M2 (pro-tumoral and pro-repair) macrophages, resting mast cells, resting DCs, CD8+ T cells, T-regs, and plasma cells [11].

The infiltration of dysfunctional and exhausted CD8+ T cells and the rise in pro-tumoral chemokines and cytokines in addition to inhibitory immunological check point signaling molecules are common characteristics of both the suppressive milieu of cancer and pharmacologically treated psoriasis.

### 3.2. TNF

Many studies have demonstrated that the skin of individuals suffering from psoriasis exhibits notably elevated levels of Tumor necrosis factor TNF-α in comparison to the skin of healthy participants [12,13].

Locally produced by keratinocytes, TNF-α has the ability to activate Nuclear Factor kB (NF-kB), the gene master regulator of cytokines transcription.

TNF was first identified for its capacity to induce tumor necrosis and to promote endothelial cell death. TNF probably stimulates T-regs and MDSCs during chronic inflammation, helping in immune evasion and tumor growth. TNF is involved in both the progression of tumors and their resistance to immunotherapies, despite its function being to prevent tumor growth. TNF raises the levels of the immunosuppressive molecules TIM-3 and PD-L1, as well as the T-regs/CD8+ ratio [14].

Numerous theories have been proposed to account for TNF’s pro- and anticancer effects. It was proposed that whereas low TNF levels support the growth of cancer, high TNF levels are antitumoral [15].

TNF production has been shown in preclinical investigations to cause de-differentiation processes, which in turn cause immunogenicity reduction, tumor relapse, and loss of melanocytic markers. TNF contribution in the epithelial-to-mesenchymal transition (EMT) has also been reported in several cancer models, including renal, lung, and breast cell carcinoma [16].

Anti-TNF medications are typically not used for people who have a high risk of cancer since they seem to be more immunosuppressive. There are now five FDA-approved TNF-inhibitors: etanercept, infliximab, adalimumab, certolizumab pegol, and golimumab.

The association between TNF antagonists and cancer incidence was examined in several trials; despite variations in endpoint definitions, duration of exposure, and indications, no appreciably elevated risk of cancer was found. However, many studies revealed a statistically significant elevated risk for non-melanoma skin cancers (NMSC) and lymphoproliferative disorders [14,17,18,19].

### 3.3. IL-17 Microenvironment and Psoriasis

Interleukin-17 is a proinflammatory cytokine, produced mainly by CD4+ T cells, in particular, T helper 17 (Th17) cells [20].

Th17 differentiation is triggered by cytokines such as IL-6, IL-1b, TGF-β, and IL-21. Through a variety of mechanisms, including the direct recruitment of MDSC, the synthesis of GM-CSF in oncogene-driven cancer cells, the stimulation of the epithelial-mesenchymal transition, and the production of VEFG and chemokines (such as CXCL12 and CXCR4), IL-17 exerts a pro-tumorigenic effect.

Tumor fibroblasts and antigen-presenting cells (APC) promote Th17 expansion and inhibit TME immunity through a feedback loop [20].

The same feed-forward stimuli are observed in psoriasis, where TNF-α, IL-26, and IL-29 are induced by IL-17, resulting in maintaining barrier dysfunction and dysregulation [21]. By causing epidermal hyperplasia, controlling keratinocytes proliferation, and regulating leukocyte subsets into the skin, these signals increase keratinocyte-derived inflammation and drive the development of mature psoriatic plaques. These signals also create a feed-forward inflammatory response that activates other proliferative pathways, such as the MAPK cascade and signals transducers and activators of transcription STAT.

Together with TNF, IL-17 also works in concert to enhance the transcription of many proinflammatory genes (TNF, IL-1β, IL-6, and IL-8), which in turn stimulates MDSC and promotes the development of Th17 cells in the skin and draining lymph nodes [22].

Through the secretion of G-CSF, Th17 cells have been demonstrated to stimulate the proliferation of tumor-associated neutrophils in preclinical models. IL-17 also modulates NK activities, as NK enrichment has been observed in lungs, bowel, and skin cancer (Crosby & Kronenberg, 2018).

Tissue-specific niches influence IL-17 responses; for example, IL-17 blockage reduces carcinogenesis and proliferation in colon cancer. Th17 stimulates neutrophil recruitment in lung cancer, which results in an immunosuppressive TME. In skin cancer, IL-17 activates epidermal growth factor receptor (EGFR) and STAT 3 increasing the tumor growth. [23,24].

Furthermore, elevated IL-17 levels are associated with both antiangiogenic resistance and chemoresistance [23,24].

Anti-IL-17 became a disease-specific target as a result of more accurate knowledge of the biology of both cancer and psoriasis, with the goals of more effective treatment and a reduction in the occurrence of major adverse events.

However, in practical practice, anti-IL-17 medications are exclusively used to treat psoriasis; anti-IL-17 therapies have not been approved for cancer treatment, most likely due to the pleiotropic effect of IL-17 and the intricate mechanisms of interaction with the TME.

As far as we know, based on research on clinicaltrials.gov, there are not any active clinical trials utilizing anti-IL-17 combos with anticancer treatments.

### 3.4. IL-23 Microenvironment and Psoriasis

Interleukin-23 is a proinflammatory cytokine that belongs to the IL-12 family of cytokines. It is released by activated APC, including macrophages and dendritic cells, as IL-12 and IL-27. It may be the origin of many autoimmune responses because of its capacity to promote the growth of T helper type 17 (Th17) cells and to enhance interferon (IFN -γ) production [25,26].

It has pleiotropic effects that are both pro- and antitumoral. Fibrosarcoma and cutaneous papilloma incidence were lower in IL-23-deficient animals, and this tumor resistance was connected with a significant rise in CD8+ T cells. Nonetheless, in TME, IL-23 inhibits CD8+ T cells’ and NK’s antitumor activity [27].

It was discovered that many malignancies overexpressed IL-23. Myeloid cells’ reactions to endogenous or external stimuli, such as hormones and damage-associated molecular patterns (DAMPs), are the main sources of IL-23 [28].

Through its interaction with the downstream pathway, IL-23 can activate proangiogenic factors (VEGF and MM9) and inflammatory cytokines (IL-22, IL-10, and IL-17) that have pro-tumoral effects [28].

Remarkably, some authors demonstrated that the responsiveness to immune checkpoint inhibitors increases with the ratio of IL-12/IL-23.

The rapid remission of psoriasis-related clinical and histologic features facilitated by biologics against IL-23 is comparable to, or superior to, that observed with IL-17 inhibition.

IL-23 is involved in the survival of Trm cells because it regulates the expression of IL-17A, which is expressed by most Trm in healed psoriatic skin. Recent studies indicate that in psoriasis, IL-23 is required for the survival of Th17 and Tc17 cells, which are effector T cells that produce IL-17A. Remarkably, Mehta et al. showed that whereas guselkumab, an IL-23 inhibitor, decreased Trm levels in healed psoriatic skin six months after beginning treatment, secukinumab, an IL-17A inhibitor, had no effect on Trm. These findings may clarify why, in contrast to other classes of psoriasis medications, IL-23 inhibitors have been linked to prolonged remissions after drug discontinuation.

According to Nast et al., infliximab, all anti-IL-17 (ixekizumab, secukinumab, bimekizumab, and brodalumab), and anti-IL23 drugs (risankizumab and guselkumab, but not tidrakizumab), were superior to ustekinumab in terms of reaching Psoriasis Area Severity Index (PASI) 90 [29].

### 3.5. Treatment Monitoring Recommendation and Specific Comorbidity

According to the EuroGuiDerm on systematic treatment of Psoriasis vulgaris, patients were identified as candidates for topical therapy or systematic therapy in accordance with each country’s national disease severity grading. When selecting a systemic treatment, these guidelines advise considering individual patient variables and comorbidities, pharmacological efficacy and safety, and time before beginning of therapeutic response. According to EuroGuiDerm guideline part 2, the choice of treatment for patients with cancer should include consideration of different scenarios including the risk of recurrence [29]. It is advised to start treating patients with “conventional” systemic drugs (methotrexate, acitretin, ciclosporin, and fumarates) as first-line therapy [30]. If traditional systemic medications were not tolerated, did not respond appropriately, or were contraindicated, they advise starting a biologic. Furthermore, if an oral treatment or traditional systemic drugs have proven insufficient in response or safety, the use of apremilast is recommended [30]. Apremilast is a small molecule that is authorized for use in cancer patients. Case studies have demonstrated the drug’s efficacy in treating psoriasis without increasing the risk of solid cancer recurrence [31,32].

## 4. Discussion

### 4.1. Can We Treat with Biologics Patients with a History of Malignancies?

The higher risk of cancer in the psoriasis population can be explained by a number of factors including: first, people who smoke, are obese, or have dyslipidemic syndrome are more likely to have psoriasis; second, medications used to treat it, such as methotrexate, cyclosporin, phototherapy, etc., are linked to an increased risk of cancer; and third, the main factor causing immune system impairment that promotes the development of cancer is chronic inflammation.

According to a 2013 systematic review and meta-analysis, individuals with psoriasis had an increased risk of developing lymphoma, NMSC, and some solid malignancies. However, the authors noted that there was significant heterogeneity in the studies assessing cancer risk in psoriasis patients, preventing them from being included in a meta-analysis. They proved a slightly elevated risk of some solid malignancies in psoriasis patients, particularly those associated with alcohol consumption and cigarette smoking. Furthermore, the incidence of non-melanoma skin malignancies, particularly squamous cell carcinoma, was elevated not by biologics but rather by prior exposure to ciclosporin, 8-methoxypsoralen-ultraviolet-A (PU-VA), and potentially methotrexate [33].

Since then, numerous larger studies have been undertaken to investigate the link between psoriasis and cancer.

Table 2 and Table 3 provide an overview of real-world clinical experience with biologics (Table 2) and meta-analysis (Table 3).

Recently, Vaengebjerg et al. reported a systematic review and meta-analysis of 112 studies including more than two million patients. Researchers observed that people with psoriasis had a slightly higher risk of cancer overall [44]. Additionally, they discovered a statistically significant elevated risk for NMSC and lymphoproliferative malignancies [17,18,19].

During a 52-week follow-up period, secukinumab demonstrated a favorable safety profile in patients with moderate-to-severe plaque psoriasis, according to a pooled analysis of 10 studies (4 Phase II and 6 Phase III). The incidence of infections that required antimicrobial treatment during the first 12 weeks was similar in the groups receiving secukinumab 300 mg, 150 mg, and etanercept (11.1%, 9.0%, and 9.9%, respectively), and these rates were numerically higher than those in the group receiving a placebo (7.4%) [40].

A Korean study of 191978 patients treated with either an IL-12/23 inhibitor or a TNF-α inhibitor examined the risk of cancers. The results showed that TNF-α inhibitor therapy was linked to a significantly higher risk of lymphoma and overall cancer, but IL-12/23 inhibitors were not linked to an increased risk of any cancer [48]

According to the findings of a meta-analysis, the standardized incidence ratio (SIR) for all cancers, excluding non-small cell lung cancer (NMSC), was 1.16 (95% confidence interval [CI] 1.07–1.25). Notably, psoriasis patients are more likely to develop basal cell carcinoma (BCC) (SIR = 2.00; 95% CI 1.83–2.20) and squamous cell carcinoma (SCC) (SIR = 5.3; 95% CI 2.63–10.71), but not melanoma (SIR = 1.07; 95% CI 0.85–1.35). In the pooled safety analysis, there were no reported side effects for patients receiving etanercept, and the incidence of malignant or unidentified tumors within the first 12 weeks was comparable between the secukinumab 300 mg and 150 mg and placebo groups [51].

Over 52 weeks, the exposure-adjusted incidence rates of all NMSC were 0.43/100 subject-years (5 cases) in the secukinumab 300-mg group and 0.61/100 subject-years (7 cases) in the secukinumab 150-mg group; there were no cases of NMSC with etanercept. No lymphoma was reported [51].

Bellinato et al. reported that out of 12 patients treated with anti-IL-17 (from the University Hospital of Verona), progressive disease was found in 2 patients with a cancer history. In the literature review, 10 patients from 5 articles (out of 661) were considered, none reported recurrence [38].

There has been no reported impact of biologics on HPV-related malignancy and cervical neoplasia. No incremental risk of disease recurrence or progression, nor cancer-treatment adverse effects were reported for melanoma; a reduction in the number of melanocytic nevi has been reported after secukinumab [52].

Anti-IL-17 and IL-23 were not associated with increased risk of Kaposi Sarcoma [53].

This research suggests that biologic treatments are not prohibited for patients who have previously had neoplasms and that they should be evaluated in conjunction with an evaluation of the patient’s risk of cancer recurrence (based on biomolecular factors and the stage of the tumor, nodes, and metastases). Drugs eligibility differs unfortunately in different countries as well as the availability of biomolecular assay and genetic tools for cancer prognosis assessment.

### 4.2. Is There a Timing?

As mentioned earlier and outlined in the tables, establishing a definitive safe timing remains inconclusive. This is attributed not only to the heterogeneous nature of the case reports but also to the inclusion of various neoplasms exhibiting a diverse range of prognoses. Furthermore, the patients in these series underwent treatment at different stages of their oncological journey, spanning periods of 1, 2, 3, and 10 years post-diagnosis. The absence of stratification based on the primary site or treatment received, owing to the limited size of the case series, adds complexity to drawing definitive conclusions [29].

Although it is usually accepted that biologics can be used five years following cancer diagnosis, there is not much evidence to support this timing. It is well known that there is little chance of a local or distant recurrence of cancer five years following diagnosis; this knowledge has been applied to dermatology practice as sound advice. Modern biomolecular characteristics of disease, rather than a disease-free interval and the health of the patient, must be considered.

Cancer is a multifaceted medical condition; even among patients with the same histological type and stage, several factors, such as age, gender, smoking status, past medical history, and weight, can alter the prognosis.

### 4.3. Can We Treat Psoriatic Patients during Anticancer Treatment? Are There Drugs More Involved in Psoriasis Exacerbations?

Chemotherapy exerts an immune influence by stimulating antigen relapse and inducing the death of immune cells. It also impacts the tumor microenvironment and modifies immune cells through the initiation of a temporary depletion of lymphocytes followed by a subsequent replenishment of immune cell reservoirs. These effects contribute to the maturation of dendritic cells, ultimately leading to the induction of potent antitumor responses [54].

The diverse immune effects observed in psoriasis may explain the range of clinical responses, varying from beneficial outcomes during anticancer treatments to pronounced exacerbations. Notably, like traditional chemotherapeutic agents, certain targeted agents exhibit immunomodulatory properties. They regulate the expression of Indoleamine 2,3-dioxygenase (IDO) in myeloid cells, mitigate the immunosuppressive activities of regulatory T cells (T-reg) and myeloid-derived suppressor cells (MDSCs), hinder angiogenic processes through direct effects on vascular cell proliferation, and indirectly influence growth factor production. Additionally, these agents enhance antigen presentation and promote immunogenic cell death [54].

Among the chemotherapy families, topoisomerase inhibitors (e.g., etoposide, mitoxantrone, and doxorubicin) and antimicrotubule agents (e.g., vinblastine, paclitaxel, and docetaxel) induce marginal dendritic cell death, resulting in immune stimulation. Moreover, paclitaxel increases the effector/regulatory CD8 ratio, while docetaxel increases the anti-tumor macrophage population. Cisplatin, carboplatin, and oxaliplatin increase antigen presentation and immunogenic cell death. Doxorubicin, cyclophosphamide, vinblastine, and vincristine improve DC function [54,55].

No recommendations are available regarding combinations of anti-psoriasis and target therapies (e.g., multi-kinase tyrosin kinase inhibitors, anti-angiogenetic antibodies).

Several immune checkpoint inhibitors—anti-cytotoxic T-lymphocyte-associated protein 4 (CTLA4), anti-programmed death (PD1), and anti-PDL1—are approved for the treatment of solid tumors. Indeed, a number of cancers, including melanoma head and neck cancer, cutaneous squamous cell carcinoma, lung, kidney, and breast cancer, can pre-sent overexpression of the PD-1 ligand (PD-L1) by the tumor cells as a mechanism of immune evasion. These medications function by obstructing critical steps in the immune cascade. Specifically, anti-PD1 drugs (Nivolumab, Dostarlimab, Pembrolizumab, and Cemiplimab) hinder the PD-1 receptor, responsible for inducing T-cell apoptosis, thus preventing excessive proliferation and function. On the other hand, anti-PDL1 drugs (atezolizumab, avelumab, and durvalumab) impede the ligand, inhibiting the activation of the signaling cascade. This inhibition consequently eliminates the brake on the immune response.

Conversely, experiences with psoriasis outcomes during treatment are abundant, though often based on limited data and few follow-ups. In Table 4, we document our encounters with oncological treatments.

Exacerbation of psoriasis is a frequently observed occurrence during anti-PD-1 or anti-PD-L1 therapies. Peled P demonstrated reduced expression of the PD-1 receptor in peripheral blood lymphocytes among patients with psoriatic arthritis and rheumatoid arthritis. This reduction seems to exhibit an inverse correlation with the severity of joint disease, potentially decreasing the lymphocyte population susceptible to inhibition by the PD-1 pathway [60].

Effectively managing these exacerbations is crucial for enabling the continuation of treatment.

A high Psoriasis Area and Severity Index (PASI) score has a detrimental impact on the patient’s quality of life and adherence to therapies [52]. In recent studies, the phosphodiesterase (PDE) inhibitor apremilast has been employed in psoriasis induced by checkpoint inhibitors. This drug down-regulates pro-inflammatory cytokines/chemokines, including TNF-alpha, IFN-gamma, IL-23, IL-12, and IL6, while increasing anti-inflammatory cytokines such as IL-10, IL-13, and IL4 [61].

Dulos et al. demonstrated that PD1 axis blockade affects Th1 and Th17 signaling pathways, leading to the overexpression of proinflammatory cytokines, including IL-17 and IL-22 [62]. Typically, concurrent treatment with acitretin is prescribed as the initial medication. If regression occurs with antipsoriasis treatment, immunotherapy for cancer can usually be continued, and further outbreaks of psoriasis are rarely reported [33,56,58,59,63].

Boningen et al. in an analysis of 21 patients treated with immunotherapy reported a mean time of onset between anti-PD-1 treatment and psoriasis flare of 50 days, with an average of 90 days for de novo cases and 33 days for pre-existing conditions [57]. Numerous other experiences have supported and confirmed this evidence reported [33,56,58,59,63].

A multidisciplinary international panel has undertaken the task of addressing the management of individuals with previously treated solid tumors. The consensus among the authors is that, before contemplating new therapies for psoriasis, it is crucial to consider the prognosis of the cancer. In cases where patients have a favorable cancer prognosis, the outcomes when treated with systemic psoriasis therapies are expected to be akin to those of non-treated individuals with solid tumors. Conversely, for patients with a less favorable cancer prognosis, the primary objective shifts to enhancing their quality of life. In such instances, the potential benefits of treating psoriasis may outweigh the theoretical risks associated with cancer progression [64].

## 5. Conclusions

Owing to a lack of guidelines, a multidisciplinary approach is necessary. When determining how to treat a patient, the clinical severity of psoriasis, cancer stage and prognosis, and the patient’s wishes should all be considered. Decisions should be made jointly with the patient using a personalized approach. When treating psoriasis, IL-17 and IL-23 inhibitors are typically recommended. Before starting immunotherapy for neoplasms, dermatology should be consulted for assessment and management of any patient with a positive history of immune-mediated skin disorders.

We suggest collaboration between dermatologists and oncologists, as this coordination is lacking in Italy, at least in smaller hospitals. Notwithstanding the rarity of patients with both conditions (psoriasis and cancer), multidisciplinary discussion of the benefits and drawbacks of anti-psoriasis and anti-cancer treatments are not routinely held, even in tertiary centers. Patients are frequently followed for their different pathologies (cardiovascular and neurological illness, metabolic disorders, etc.) in different clinics and even different locations.

The strength of our paper is the multidisciplinary contribution (both dermatologists and oncologists) to this highly contentious topic; its limitation, however, is that it is not a systematic study.

No timing limit is confirmed. The management approach for psoriasis in the context of cancer depends on the severity of psoriasis and the aggressiveness of the underlying cancer. In cases where the cancer is non-aggressive (i.e., disease is growing slowly and does not pose an immediate threat), severe psoriasis may be addressed even after the cancer has been excised. However, when dealing with a highly aggressive cancer and moderate psoriasis, treatment with topical therapies is recommended.

Collecting prospective data on interactions and outcomes, alongside substantial retrospective information, is essential for understanding the implications of using biologics concurrently with anti-cancer therapy and comprehending the dermatologic effects of diverse classes of antitumoral cancer treatments. This approach is crucial for informing optimal decision-making in the complex scenarios where psoriasis and cancer coexist.

## Figures and Tables

**Figure 1 ijms-24-17540-f001:**
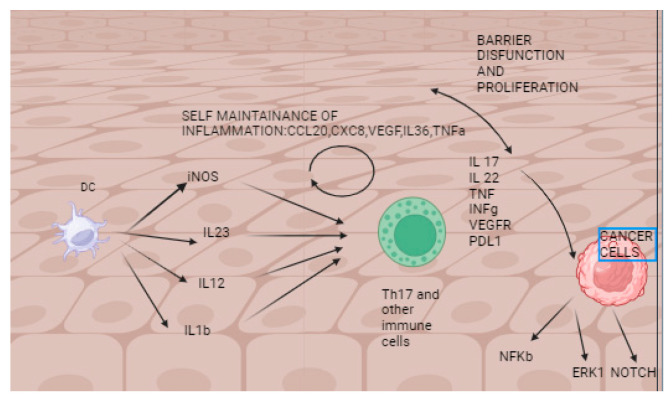
Dendritic cells (DC) start the immunological cascade in psoriasis and produce IL-1b, IL-12, and IL-23. The most crucial cytokine pathway is the IL-17/IL-23 axis. IL-17, IL-21, and IL-22 cause epidermal barrier dysfunction and keratinocyte proliferation. IL-17 stimulates many proliferative pathways, such as NOTCH, ERK1, and NF-Kb.

**Table 1 ijms-24-17540-t001:** Biotechnological anti-interleukin treatments.

Biological Drug	Mechanism of Action	FDA Approval for Psoriasis
Etanercept	Anti-TNFα	2004
Infliximab	Anti-TNFα	2006
Adalimumab	Anti-TNFα	2008
Certolizumab-pegol	Anti-TNFα	2008
Golimumab	Anti-TNFα	2009
Ustekinumab	Anti p40 IL-12/IL-23	2009
Secukinumab	Anti-IL-17A	2015
Ixekizumab	Anti-IL-17A	2016
Brodalumab	Anti-IL-17RA	2017
Guselkumab	Anti-p19 IL-23	2017
Tildrakizumab	Anti-p19 IL-23	2018
Risankizumab	Anti-IL-23A	2022
Bimekizumab	IL-17A/F	2023

Legend: FDA: Food and Drug Administration.

**Table 2 ijms-24-17540-t002:** Real world experience with Biologics.

Studies	N° pts	Drugs	Timing from Diagnosis to Treatment	Recurrence or Progression	Follow-Up Duration	New Primary Tumors
Odorici G [34]	14	Anti-TNFα, Anti-IL-12/23, anti-IL-17; anti-IL-23.	3.6 y	No recurrence or progression	34 months	(1 SCLC, 1 multiple myeloma, 1 NMSC)
Kahn JS [35]	16	Apremilast, Anti-TNFα, Anti-IL-12/23, anti-IL-17; anti IL-23.	4.7 y	No recurrence or progression	Retrospective	None
Mastorino L [36]	37	Anti-TNFα, Anti-IL-12/23, anti-IL-17; anti-IL-23.	7 y	No progression. 1 endometrial cancer recurrence	33 months	None
Mastorino L [37]	7	Anti-IL-23,	3–20 y	No recurrence or progression	11–15 months	1 NMSC
Bellinato F [38]	12	Anti—IL-17	15 m	No recurrence, 2 disease progression	46 months	None
Valenti M [39]	16	Anti-TNFα, Anti-IL-12/23, anti-IL-17; anti-IL-23.	5–10 y	No recurrence	22 months	None
Pellegrini C [40]	42	Anti-IL-17	3.5 y	No recurrence, 2 progressions in metastatic disease	12 months	(1 SCLC, 1 1 breast cancer, 1 glioblastoma)

**Table 3 ijms-24-17540-t003:** Summarizes effects on recurrence and progression in the recent meta-analyses.

Studies	N° pts	Drugs	Recurrence or Progression	New Primary Tumors
Augustin M [41]	1756	Secukinumab	No recurrence or progression.	NR
Lebwohl M [42]	10,685	Secukinumab	No recurrence or progression.	EAIR of malignancy was 0.85/100 PTY, corresponding to 204 patients per 23 908 PY
Blauvelt A [43]	1721	Guselkumab	No recurrence or progression.	EAIR of malignancy was 0.74/100 PY, corresponding to 53 patients per 7117 PY
[44]	112 studies including 2,053,932 patients	Patients receiving biologic therapy and non-biologic therapy	No recurrence or progression.	No increase in cancer was seen among patients with psoriasis treated with biologic agents
Gupta A [45]	31 studies, 24,328 persons	Patients receiving biologic therapy and non-biologic therapy	Rates of cancer recurrence were similar among individuals not on immunosuppression, receiving an anti-TNF, immunomodulators, or combination immunosuppression. Patients receiving ustekinumab and vedolizumab had lower rates of cancer.	NR
Smith SD [46]	17 studies, including 6892 patients	Ixekizumab		55 patients developed NMSC, 1 melanoma, 1 dermatofibrosarcoma protuberans
Gottlieb A 2022 [47]	8891	Secukinumab	EAIR/100PY 224 (PsO) EAIR/100PY 159.2 (PsA) EAIR/100PY 125.5 (AS)	NR
Jung JM [48]	191,678	TNF-α inhibitor/IL12/23 inhibitor	SIR, 1.12; 95% (CI 1.09–1.14). TNF-α inhibitor (aHR 1.41; 95% CI 1.01–1.97). IL-12/23 inhibitor (aHR, 0.57; 95% CI 0.37–0.87).	NR
Gargiulo L 2023 [49]	606	Brodalumab	The log-rank test and Cox regression did not detect any differences in drug survival regarding BMI classes, comorbidities, involvement of difficult-to-treat areas, and previous exposure to biologics	1 prostate cancer 1 breast cancer, (after 16 weeks of treatment) they both stopped brodalumab to undergo surgery
Hellgren K [50]	55,850	TNF-α inhibitor	HR for solid cancer overall was 1.0 (0.9–1.2) for TNFi-exposed vs biologics-naïve PsA	NR

Abbreviations: EAIR Exposed-adjusted incidence rates; PY patient year; NR not reported; SIR standardized incidence rates; CI confidence interval; aHR adjusted hazard ratio; PsO Psoriasis; PsA psoriatic arthritis; AS ankylosing spondylitis.

**Table 4 ijms-24-17540-t004:** Experiences of oncological treatment while on biologics for psoriasis.

Study	Pts/Biologics	Prior Diagnosis of Neoplasm ≥ 5 y	Prior Diagnosis of Neoplasm ≥ 10 y	Neoplasm after Biologics	Biologics Start after Prior Neoplasm	Biologics Start during Anti Neoplastic Treatment	Effect
Vodouri D [56]	5 (P)	Y	N	N	N	Y	Exacerbations
Bonigen J [57]	21 (P)	Y	N	N	N	Y	De novo 90 d pre-existing 33 d
Johnson D [58]	1 (P)	N	N	N	N	Y	Stop pembrolizumab for psoriasis started sekukinumab succumbed
Esfahani K [59]	1 (P)	Y	N	N	Y	Y	Stop pembrolizumab progression

Abbreviation: Y = yes; N = no P psoriasis; PsA psoriatic arthritis; AS = ankylosing spondylitis; EAIR = exposure-adjusted incidence rates.

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
