# Peer review of "A Multidisciplinary Approach to Patients with Psoriasis and a History of Malignancies or On-Treatment for Solid Tumors: A Narrative Literature Review"

_ijms, 2023, doi:10.3390/ijms242417540_

Round 1

Reviewer 1 Report

Comments and Suggestions for Authors

The authors reported the results of a narrative review aiming to provide an overview on the use of biologics in patients with a history of malignancy. The manuscript is interesting ad well written. The topic is original and relevant in the field as updated data are needed.  References are appropriate. Tables and figures improve the quality of the paper. The manuscript is suitable for publication after some changes.

My comments:

- Abstract: please rewrite the aim of the study. It is unclear.

- Keywords: there are ";" highlighted. Please correct.

- Introduction: Sentence 1 lacks reference. Please add.

- Introduction: current treatment scenario for psoriasis patients should be discussed (doi:10.1111/jdv.16915 and doi:10.1111/jdv.16926)

- Methods: this section is lacking. How have you performed the review? Please specify

- Discussion: current guidelines suggest using biologic at least 5 years after cancer remission. Please discuss

- Discussion: please add strengths and limitations of your study

- Discussion: please discuss the role of apremilast which currently is the only "biologic" drug approved for cancer patients

Author Response

Thanks for allowing us to submit a revised draft of the manuscript. We earnestly appreciate the time and effort that you have dedicated to providing your valuable feedback on our manuscript. We are grateful to the Reviewers for their insightful comments. We have incorporated changes to reflect the suggestions provided by the Reviewers. Herein, we report a point-by-point response to reviewers, and we have highlighted the changes in yellow within the marked revised manuscript.

- Abstract: please rewrite the aim of the study. It is unclear.

We clarified it adding: we performed a narrative review of literature aiming to provide practical tips to oncologist and dermatologists,to suggest when interrupting and/or starting biologics for psoriasis.

- Keywords: there are ";" highlighted. Please correct.

We corrected

- Introduction: Sentence 1 lacks reference. Please add.

We added the reference

- Introduction: current treatment scenario for psoriasis patients should be discussed (doi:10.1111/jdv.16915 and doi:10.1111/jdv.16926)

we inserted as requested:

According with the Euroguiderm on systematic treatment of Psoriasis vulgaris patients were classified as candidate for topical therapy or systematic therapy following national disease severity grading of each country.

These guidelines recommend to consider individual patient factors and comorbidities, efficacy ad safety, time until onset of treatment response when choosing a systemic treatment. In patients with moderate to severe psoriasis a systemic treatment is required. The initiation of “conventional” systemic agents (acitretin,ciclosporin,fumarates and methotrexate) as first line treatment is suggested..[Nast A, 2020 ]

They recommend the initiation of a biologic if conventional systemic agents were inadeguate in response or contraindicated or not tolerated.

Moreover, the use of apremilast is preferable if an oral treatment or conventional systemic agents were inadeguate in response or safety..[Nast A, 2020 ]

According to EuroGuiDerm guideline part 2  the choice of treatment in patients with cancer should consider different scenarios including the risk of recurrence .[Nast A, 2021 ]

  At drug level,infliximab, all of the anti IL17 drugs ( ixekizumab,secukinumab,bimekizumab and brodalumab) and anti IL23 drugs(risankizumab and guselkumab,but not tidrakizumab)were superior than ustekinumab  in term of reaching Psoriasis area severity index (PASI) 90.[Nast A,2021 ]. 

- Methods: this section is lacking. How have you performed the review? Please specify

We performed a research on English written papers published from 2000 to 2023, utilizing the PubMed database. The terms "carcinoma" and/or "history of malignancies” and “psoriasis," along with subheadings such as pathogenesis, biologics, anti TNF, anti IL17, anti IL23, and cancer treatment and chemotherapy were systematically searched . Inclusion criteria comprised (1) case series or case reports specifically addressing cancer risk in patients with psoriasis, and biologic treatment of psoriasis and the risk of malignancy or cancer recurrence (2) review studies, meta-analyses, and systematic reviews focused on effect of biologics in cancer incidence in patients with psoriasis who have had a prior malignancy. Exclusion criteria included (1) papers written in languages other than English,. In addition, chosen pieces from international meetings were used to enhance the discussion of our review.

- Discussion: current guidelines suggest using biologic at least 5 years after cancer remission. Please discuss

We discuss as requested

Although is widely accepted that biologic can be used 5 years after cancer remission, this timing is not supported by significant evidence. It is commonly recognised that after 5 year from cancer diagnosis, the risk of local or distant recurrence is low and this has been translated in dermatology practice as a good advice. However, up to date biomolecular features of disease more than disease free interval and patients’ conditions must be considered.

- Discussion: please add strengths and limitations of your study

                                           In our opinion the strength of this narrative review of the literature is the multidisciplinary contribution (oncologists and dermatologists) to each debated point, the limitation is to be not a systematic review.

- Discussion: please discuss the role of apremilast which currently is the only "biologic" drug approved for cancer patients

we add as requested:

Apremilast belongs to small molecules and is approved for cancer patients and case series showed that the drug is effective in managing psoriasis without causing recurrence of solid cancer.[Gambardella A 2022, Apalla Z 2019]

Reviewer 2 Report

Comments and Suggestions for Authors

The review focuses on the need of treating Psoriasis in patients with neoplasm. Or rather the question if it should be treated or not. Over all the authors provide excellent background on how the treatment process for reducing the non-melanoma is increasing the risk of Psoriasis in these patients. 

1. Its my opinion that too much information regarding the immunological pathway disrupted in Psoriasis is given. I think some of the introduction can be reduced or modified to support the discussion. The most important aspect of the article is given in discussion and conclusion. Its my suggestion that some of introduction section could be modified to corelate with the discussion. 

2. In final conclusion the authors make a recommendation that a oncologist and dermatologist need to co-ordinate the treatment options for patients that have neoplasm and develop Psoriasis during the treatment process of the said neoplasm. Is there an evidence that this co-ordination does not exist? Also, the neoplasm under discussion is non-melanoma cancers which are rare and in most cases diagnosed by a dermatologist. Is it possible given their small number these case studies are not tracked well?

Author Response

Thanks for allowing us to submit a revised draft of the manuscript. We earnestly appreciate the time and effort that you have dedicated to providing your valuable feedback on our manuscript. We are grateful to the Reviewers for their insightful comments. We have incorporated changes to reflect the suggestions provided by the Reviewers. Herein, we report a point-by-point response to reviewers, and we have highlighted the changes in yellow within the marked revised manuscript.

The review focuses on the need of treating Psoriasis in patients with neoplasm. Or rather the question if it should be treated or not. Over all the authors provide excellent background on how the treatment process for reducing the non-melanoma is increasing the risk of Psoriasis in these patients. 

  1. Its my opinion that too much information regarding the immunological pathway disrupted in Psoriasis is given. I think some of the introduction can be reduced or modified to support the discussion. The most important aspect of the article is given in discussion and conclusion. Its my suggestion that some of introduction section could be modified to correlate with the discussion. 

Thank you for your suggestion, we have shortened the introduction chapter in favour of the conclusion chapter

  1. In final conclusion the authors make a recommendation that a oncologist and dermatologist need to co-ordinate the treatment options for patients that have neoplasm and develop Psoriasis during the treatment process of the said neoplasm. Is there an evidence that this co-ordination does not exist? Also, the neoplasm under discussion is non-melanoma cancers which are rare and in most cases diagnosed by a dermatologist. Is it possible given their small number these case studies are not tracked well?

We recommend a cooperation among dermatologists and oncologists, because in Italy at least in small Hospitals these coordination does not exist. Although patients with concurrent cancer and psoriasis are not common a meeting “on demand” to evaluate advantages and disadvantages of anti-psoriasis and/or anti-cancer treatments is not standardized even in tertiary hospitals. Patients are frequently followed for theirdifferent comorbidities in different clinics and even different locations.

Reviewer 3 Report

Comments and Suggestions for Authors

Dear authors,

I read your review concerning a multidisciplinary approach in patients with psoriasis and a history of malignancies or on treatment for solid tumors. The title is promising but the paper looks like a commentary more than a review. At the same time, the language is informal and not adequate for a publication. It is hard to correct the paper: no APA style was respected; the text is unclear and not formatted. In this view, I can’t provide a helpful manuscript review. I suggest providing a better version of the manuscript to be corrected.

1.     The abstract is general and doesn’t report any significant information. Moreover, Line 21, pros and cons? Explain.

2.     Keywords, form the font.

3.     Lines 47-55 are too superficial. I suggest reading and citing:

-        Caputo V, Strafella C, Cosio T, Lanna C, Campione E, Novelli G, Giardina E, Cascella R. Pharmacogenomics: An Update on Biologics and Small-Molecule Drugs in the Treatment of Psoriasis. Genes (Basel). 2021 Sep 10;12(9):1398. doi: 10.3390/genes12091398. PMID: 34573380; PMCID: PMC8470543.

-        Jung JM, Kim YJ, Chang SE, Lee MW, Won CH, Lee WJ. Cancer risks in patients with psoriasis administered biologics therapy: a nationwide population-based study. J Cancer Res Clin Oncol. 2023 Sep 27. doi: 10.1007/s00432-023-05387-6. Epub ahead of print. PMID: 37755577.

-        Papp KA, Melosky B, Sehdev S, Hotte SJ, Beecker JR, Kirchhof MG, Turchin I, Dutz JP, Gooderham MJ, Gniadecki R, Hong CH, Lambert J, Lynde CW, Prajapati VH, Vender RB. Correction to: Use of Systemic Therapies for Treatment of Psoriasis in Patients with a History of Treated Solid Tumours: Inference-Based Guidance from a Multidisciplinary Expert Panel. Dermatol Ther (Heidelb). 2023 Aug;13(8):1889-1890. doi: 10.1007/s13555-023-00949-5. Erratum for: Dermatol Ther (Heidelb). 2023 Apr;13(4):867-889. PMID: 37410207; PMCID: PMC10366052.

-        Hellgren K, Ballegaard C, Delcoigne B, Cordtz R, Nordström D, Aaltonen K, Gudbjornsson B, Love TJ, Aarrestad Provan S, Sexton J, Zobbe K, Kristensen LE, Askling J, Dreyer L. Risk of solid cancers overall and by subtypes in patients with psoriatic arthritis treated with TNF inhibitors - a Nordic cohort study. Rheumatology (Oxford). 2021 Aug 2;60(8):3656-3668. doi: 10.1093/rheumatology/keaa828. PMID: 33401297.

4.     Line 59, correct interleukins

5.     Line 62, full stop missing.

6.     You could use BioRender to make figures.

7.     Figure 1 should be corrected: grammatical and syntactic errors.

8.     In Table 1, you reported FDA approval, but biological drugs used in psoriasis are unavailable in the country (e.g. Mexico, Malaysia, Corea). Then, I know Ixekizuamb inhibits IL-17, but it is unclear in the table, as well as for IL-23 inhibitors. Moreover, among IL-23, you should report the IL-23/12 inhibitors.

9.     Line 96, PME refers to the cutaneous localization of the disease? This should be clarified, as clinical variants of psoriasis reflect different pathways (IL-17 or IL-36). Explain.

10.  Line 104, I think interleukin (IL) should be reported in the first lines of the text, not in the middle.

11.  Line 115, correct the text; it is not clear.

12.  Line 119: correct the text; it is inappropriate.

13.  References should be reported after the author's name in the text, not at the end of the sentence.

14.  Extensive editing of the English language is required. The text is not clear.

15. The full name should be reported at the beginning of a sentence. Not acronyms.  

16. The material and method section is missing. How did you perform the review?

17.  Line 354, correct references.

18. Limitations of the study are missing. 

Comments on the Quality of English Language

Extensive editing of English language required

Author Response

Thanks for allowing us to submit a revised draft of the manuscript. We earnestly appreciate the time and effort that you have dedicated to providing your valuable feedback on our manuscript. We are grateful to the Reviewers for their insightful comments. We have incorporated changes to reflect the suggestions provided by the Reviewers. Herein, we report a point-by-point response to reviewers, and we have highlighted the changes in yellow within the marked revised manuscript.

Dear authors,

I read your review concerning a multidisciplinary approach in patients with psoriasis and a history of malignancies or on treatment for solid tumors. The title is promising but the paper looks like a commentary more than a review. At the same time, the language is informal and not adequate for a publication. It is hard to correct the paper: no APA style was respected; the text is unclear and not formatted. In this view, I can’t provide a helpful manuscript review. I suggest providing a better version of the manuscript to be corrected.

We revised the manuscript and correct the style according to APA style.

  1. The abstract is general and doesn’t report any significant information. Moreover, Line 21, pros and cons? Explain.

We explained benefits and risks

  1. Keywords, form the font. We corrected it
  2. Lines 47-55 are too superficial. I suggest reading and citing:

-        Caputo V, Strafella C, Cosio T, Lanna C, Campione E, Novelli G, Giardina E, Cascella R. Pharmacogenomics: An Update on Biologics and Small-Molecule Drugs in the Treatment of Psoriasis. Genes (Basel). 2021 Sep 10;12(9):1398. doi: 10.3390/genes12091398. PMID: 34573380; PMCID: PMC8470543.

We inserted in the introduction

-        Jung JM, Kim YJ, Chang SE, Lee MW, Won CH, Lee WJ. Cancer risks in patients with psoriasis administered biologics therapy: a nationwide population-based study. J Cancer Res Clin Oncol. 2023 Sep 27. doi: 10.1007/s00432-023-05387-6. Epub ahead of print. PMID: 37755577.

we inserted in the analyzed studies

-        Papp KA, Melosky B, Sehdev S, Hotte SJ, Beecker JR, Kirchhof MG, Turchin I, Dutz JP, Gooderham MJ, Gniadecki R, Hong CH, Lambert J, Lynde CW, Prajapati VH, Vender RB. Correction to: Use of Systemic Therapies for Treatment of Psoriasis in Patients with a History of Treated Solid Tumours: Inference-Based Guidance from a Multidisciplinary Expert Panel. Dermatol Ther (Heidelb). 2023 Aug;13(8):1889-1890. doi: 10.1007/s13555-023-00949-5. Erratum for: Dermatol Ther (Heidelb). 2023 Apr;13(4):867-889. PMID: 37410207; PMCID: PMC10366052.

We inserted in the end of discussion

-        Hellgren K, Ballegaard C, Delcoigne B, Cordtz R, Nordström D, Aaltonen K, Gudbjornsson B, Love TJ, Aarrestad Provan S, Sexton J, Zobbe K, Kristensen LE, Askling J, Dreyer L. Risk of solid cancers overall and by subtypes in patients with psoriatic arthritis treated with TNF inhibitors - a Nordic cohort study. Rheumatology (Oxford). 2021 Aug 2;60(8):3656-3668. doi: 10.1093/rheumatology/keaa828. PMID: 33401297.

we inserted in the analyzed studies

  1. Line 59, correct interleukins. We corrected
  2. Line 62, full stop missing. We corrected.
  3. You could use BioRender to make figures. We used biorender.
  4. Figure 1 should be corrected: grammatical and syntactic errors. Corrected
  5. In Table 1, you reported FDA approval, but biological drugs used in psoriasis are unavailable in the country (e.g. Mexico, Malaysia, Corea). Then, I know Ixekizuamb inhibits IL-17, but it is unclear in the table, as well as for IL-23 inhibitors. Moreover, among IL-23, you should report the IL-23/12 inhibitors. We corrected
  6. Line 96, PME refers to the cutaneous localization of the disease? This should be clarified, as clinical variants of psoriasis reflect different pathways (IL-17 or IL-36). Explain. Yes with PME we referts to cutaneous localization of disease.
  7. Line 104, I think interleukin (IL) should be reported in the first lines of the text, not in the middle. Corrected
  8. Line 115, correct the text; it is not clear. Corrected
  9. Line 119: correct the text; it is inappropriate. Corrected
  10. References should be reported after the author's name in the text, not at the end of the sentence. We corrected according to APA style using Zotero software
  11. Extensive editing of the English language is required. The text is not clear. We revised the English language
  12. The full name should be reported at the beginning of a sentence. Not acronyms.  We corrected it
  13. The material and method section is missing. How did you perform the review? Added
  14.  Line 354, correct references. We corrected it
  15. Limitations of the study are missing.We Added

Round 2

Reviewer 3 Report

Comments and Suggestions for Authors

Dear authors,

I read your revised review concerning a multidisciplinary approach in patients with psoriasis and a history of malignancies or on treatment for solid tumors. The corrections made are not highlighted. However, the paper presents some minor points that should be addressed.

1)     You didn’t report the references suggested by the reviewer. Explain

2)     IL-17, standardized in the main text.

3)     Line 203, reference?

4)     Lines 2014, correct the reference from clinical trial. It does not respect the APA style.

5)     Line 299, correct the number format, as well as in table 3 (and the font).

6)     Lines 325-328, 441-443, unclear. Extensive English editing is required.

7)     Line 446. Definition of not aggressive?

8)     As for the previous review, the conclusion is not clear. If I read a review concerning the psoriatic treatment option in oncological patients, I would have a clear statement of how to treat the patients. If the patient has an high burden of disease both for psoriasis and neoplasms, how can I treat him? As well as if a patient has a mild psoriasis but stage IV neoplasm. This should be the real focus in the end.

9)     Data missing in the tables. Y? m? add in the captions.

10) Line 270. Medication to treat what? Cancer? psoriasis?

11) Lines 292-298, out of the aim of the review.

12) In lines 325-328 you answer at the first question. But with drugs are eligible?

Comments on the Quality of English Language

Moderate editing of English language required

Author Response

Dear Editor and dear Reviewer,

Thanks for allowing us to submit a revised draft of the manuscript. We earnestly appreciate the time and effort that you have dedicated to providing your valuable feedback on our manuscript. We are grateful to the Reviewers for their insightful comments. We have incorporated changes to reflect the suggestions provided by the Reviewers. Herein, we report a point-by-point response to reviewers, and we have highlighted the changes in yellow within the marked revised manuscript.

REVIEWER 3 Dear authors,

I read your revised review concerning a multidisciplinary approach in patients with psoriasis and a history of malignancies or on treatment for solid tumors. The corrections made are not highlighted. However, the paper presents some minor points that should be addressed.

  • You didn’t report the references suggested by the reviewer. Explain

We had inserted the references  requested in the previous submission except

Caputo V, Strafella C, Cosio T, Lanna C, Campione E, Novelli G, Giardina E, Cascella R. Pharmacogenomics: An Update on Biologics and Small-Molecule Drugs in the Treatment of Psoriasis. Genes (Basel). 2021 Sep 10;12(9):1398. doi: 10.3390/genes12091398. PMID: 34573380; PMCID: PMC8470543.

That we inserted now in the introduction

-        Jung JM, Kim YJ, Chang SE, Lee MW, Won CH, Lee WJ. Cancer risks in patients with psoriasis administered biologics therapy: a nationwide population-based study. J Cancer Res Clin Oncol. 2023 Sep 27. doi: 10.1007/s00432-023-05387-6. Epub ahead of print. PMID: 37755577.

we inserted in the analyzed studies

-        Papp KA, Melosky B, Sehdev S, Hotte SJ, Beecker JR, Kirchhof MG, Turchin I, Dutz JP, Gooderham MJ, Gniadecki R, Hong CH, Lambert J, Lynde CW, Prajapati VH, Vender RB. Correction to: Use of Systemic Therapies for Treatment of Psoriasis in Patients with a History of Treated Solid Tumours: Inference-Based Guidance from a Multidisciplinary Expert Panel. Dermatol Ther (Heidelb). 2023 Aug;13(8):1889-1890. doi: 10.1007/s13555-023-00949-5. Erratum for: Dermatol Ther (Heidelb). 2023 Apr;13(4):867-889. PMID: 37410207; PMCID: PMC10366052.

We inserted in the end of discussion

-        Hellgren K, Ballegaard C, Delcoigne B, Cordtz R, Nordström D, Aaltonen K, Gudbjornsson B, Love TJ, Aarrestad Provan S, Sexton J, Zobbe K, Kristensen LE, Askling J, Dreyer L. Risk of solid cancers overall and by subtypes in patients with psoriatic arthritis treated with TNF inhibitors - a Nordic cohort study. Rheumatology (Oxford). 2021 Aug 2;60(8):3656-3668. doi: 10.1093/rheumatology/keaa828. PMID: 33401297.

we inserted in the analyzed studies

2)     IL-17, standardized in the main text. We substituted all IL 17 with IL-17

3)     Line 203, reference? We add Kuen D et al 2020

4)     Lines 2014, correct the reference from clinical trial. It does not respect the APA style.

There are no trials  on IL-17 treatment of cancer,  we correct the sentence as follows: As far as we know, based on research on clinicaltrials.gov, there aren't any active clinical trials utilizing anti-IL-17 combos with anticancer treatments.

5)     Line 299, correct the number format, as well as in table 3 (and the font). We removed 46

6)     Lines 325-328, 441-443, unclear. Extensive English editing is required.

We revised as follows

No incremental risk of disease recurrence/progression neither cancer-treatment adverse effects were reported for melanoma patients; a reduction in the number of melanocytic nevi has been reported after secukinumab (Li et al., 2022)

Anti-IL-17 and IL-23 were not associated with increased risk of Kaposi Sarcoma (Genovese et al., 2019).

 Notwithstanding the rarity of patients with both conditions (psoriasis and cancer) multidisciplinary discussion of the benefits and drawbacks of anti-psoriasis and/or anti-cancer treatments are not routinely held, even in tertiary centers. Patients are frequently followed for their different pathologies (cardiovascular and neurological illness, metabolic disorders, etc.) in different clinics and even different locations.

The strength of our paper is the multidisciplinary contribution (both dermatologists and oncologists) to this highly contentious topic; its limitation, however, is that it is not a systematic review.

7)     Line 446. Definition of not aggressive? We corrected (disease is growing slowly and does not pose an immediate threat),

8)     As for the previous review, the conclusion is not clear. If I read a review concerning the psoriatic treatment option in oncological patients, I would have a clear statement of how to treat the patients. If the patient has an high burden of disease both for psoriasis and neoplasms, how can I treat him? As well as if a patient has a mild psoriasis but stage IV neoplasm. This should be the real focus in the end.

We appreciate your effort, however, there are no guidelines on the topic. (For example  a stage IV pancreatic cancer has a different behaviour than a stage IV thyroid cancer). So we could not provide a clear statement  on how treating the patients.

We recommend a multidisciplinary approach and “Decisions should be made jointly with the patient using a personalized approach” e.g. considering age, comorbidity, psoriasis severity, cancer prognosis.

9)     Data missing in the tables. Y? m? add in the captions.

We add  Y= yes; N= no

10) Line 270. Medication to treat what? Cancer? psoriasis?

we changed as follow   Can we treat with biologics patients with a history of malignancies?

11) Lines 292-298, out of the aim of the review.

We considered to add these part as the risk of infections may affect the anticancer treatment decision (e.g an hematological toxic drug protocol for anti-cancer treatment is not suggested for the higher risk of infection).

12) In lines 325-328 you answer at the first question. But with drugs are eligible?

We are not sure about the suggestion. We have inserted  this sentence

This research suggests that biologic treatments are not prohibited for patients who have previously had neoplasms and that they should be evaluated in conjunction with an evaluation of the patient's risk of cancer recurrence (based on biomolecular factors and the stage of the tumor, nodes, and metastases).Drugs eligibility differs unfortunately in different countries as well as the availability of biomolecular assay and genetic tools for cancer prognosis assessment.

Round 3

Reviewer 3 Report

Comments and Suggestions for Authors

Dear Authors,

All the corrections have been made and the manuscript has been improved. 

Comments on the Quality of English Language

Minor editing of English language required